**EMBO** *reports*

# ChIP-Atlas: a data-mining suite powered by full integration of public ChIP-seq data

Shinya Oki[1],[*] , Tazro Ohta[2] , Go Shioi[3], Hideki Hatanaka[4] , Osamu Ogasawara[5] , Yoshihiro Okuda[5], Hideya Kawaji[6],[7] , Ryo Nakaki[8],[9], Jun Sese[10],[11] & Chikara Meno[1],[**]

## Abstract

We have fully integrated public chromatin chromatin immunoprecipitation sequencing (ChIP-seq) and DNase-seq data (*n* > 70,000) derived from six representative model organisms (human, mouse, rat, fruit fly, nematode, and budding yeast), and have devised a data-mining platform—designated ChIP-Atlas (http://chip-atlas. org). ChIP-Atlas is able to show alignment and peak-call results for all public ChIP-seq and DNase-seq data archived in the NCBI Sequence Read Archive (SRA), which encompasses data derived from GEO, ArrayExpress, DDBJ, ENCODE, Roadmap Epigenomics, and the scientific literature. All peak-call data are integrated to visualize multiple histone modifications and binding sites of transcriptional regulators (TRs) at given genomic loci. The integrated data can be further analyzed to show TR–gene and TR–TR interactions, as well as to examine enrichment of protein binding for given multiple genomic coordinates or gene names. ChIP-Atlas is superior to other platforms in terms of data number and functionality for data mining across thousands of ChIP-seq experiments, and it provides insight into gene regulatory networks and epigenetic mechanisms.

**Keywords** ChIP-seq; data mining; DNase-seq; enhancer; transcription factor
**Subject Categories** Chromatin, Epigenetics, Genomics & Functional Genomics; Methods & Resources; Transcription

## Introduction

Chromatin immunoprecipitation sequencing (ChIP-seq) is a powerful method to analyze genome-wide binding of modified histones, RNA polymerases, and other proteins involved in transcription or regulation of gene expression such as transcription factors that recognize specific DNA sequences [1], chromatin-remodeling factors, and histone modification enzymes (collectively referred to as transcriptional regulators [TRs] in this paper). A large amount of data for both ChIP-seq and DNase-seq—a method for profiling regions of open chromatin accessible to DNase [2,3]—has been compiled by a panel of ENCODE consortia for representative model organisms (human, mouse, fruit fly, and nematode) and has served as a global resource for understanding gene regulatory mechanisms and epigenetic modifications [4–7]. In addition, several tools and databases have been developed for visualization and analysis of public ChIP-seq data in a manner largely dependent on the ENCODE project data for human and mouse [8–11]. On the other hand, a substantial amount of ChIP-seq data has been presented by various smaller projects (Fig 1A). Although such data are publicly available from Sequence Read Archive (SRA) of NCBI (https://www.ncbi.nlm.nih.gov/sra), they have been made use of to a lesser extent by the research community than have the ENCODE data for several reasons: (i) Unlike the ENCODE data, only the raw sequence data are archived in most cases, necessitating extensive bioinformatics analysis; (ii) metadata such as antigen and cell type names are often ambiguous as a result of the use of orthographic variants such as abbreviations and synonyms; and (iii) integrative analysis of such data requires skills for data mining and abundant computational resources.

Given this background, we launched a project in 2014 to fully exploit and reuse the ChIP-seq and DNase-seq data in the public domain, and we released to the public in December 2015 an easy-to-use database and associated data-mining tools that we designated ChIP-Atlas (http://chip-atlas.org; Fig EV1A). We here present the data content and features of ChIP-Atlas as compared with existing relevant tools published in the same period.

---

1   Department of Developmental Biology, Graduate School of Medical Sciences, Kyushu University, Fukuoka, Japan
2   Database Center for Life Science, Joint-Support Center for Data Science Research, Research Organization of Information and Systems, Mishima, Shizuoka, Japan
3   Genetic Engineering Team, RIKEN Center for Life Science Technologies, Kobe, Japan
4   National Bioscience Database Center, Japan Science and Technology Agency, Tokyo, Japan
5   DNA Data Bank of Japan, National Institute of Genetics, Mishima, Shizuoka, Japan
6   Preventive Medicine and Applied Genomics Unit, RIKEN Center for Integrative Medical Sciences, Kanagawa, Japan
7   RIKEN Preventive Medicine and Diagnosis Innovation Program, Saitama, Japan
8   Genome Science Division, Research Center for Advanced Science and Technology, The University of Tokyo, Tokyo, Japan
9   Rhelixa Inc., Tokyo, Japan
10  Artificial Intelligence Research Center, National Institute of Advanced Industrial Science and Technology, Tokyo, Japan
11  Humanome Lab Inc., Tokyo, Japan
    *Corresponding author. Tel: +81 92 642 6259; E-mail: soki@dev.med.kyushu-u.ac.jp
    **Corresponding author. Tel: +81 92 642 6259; E-mail: meno@dev.med.kyushu-u.ac.jp

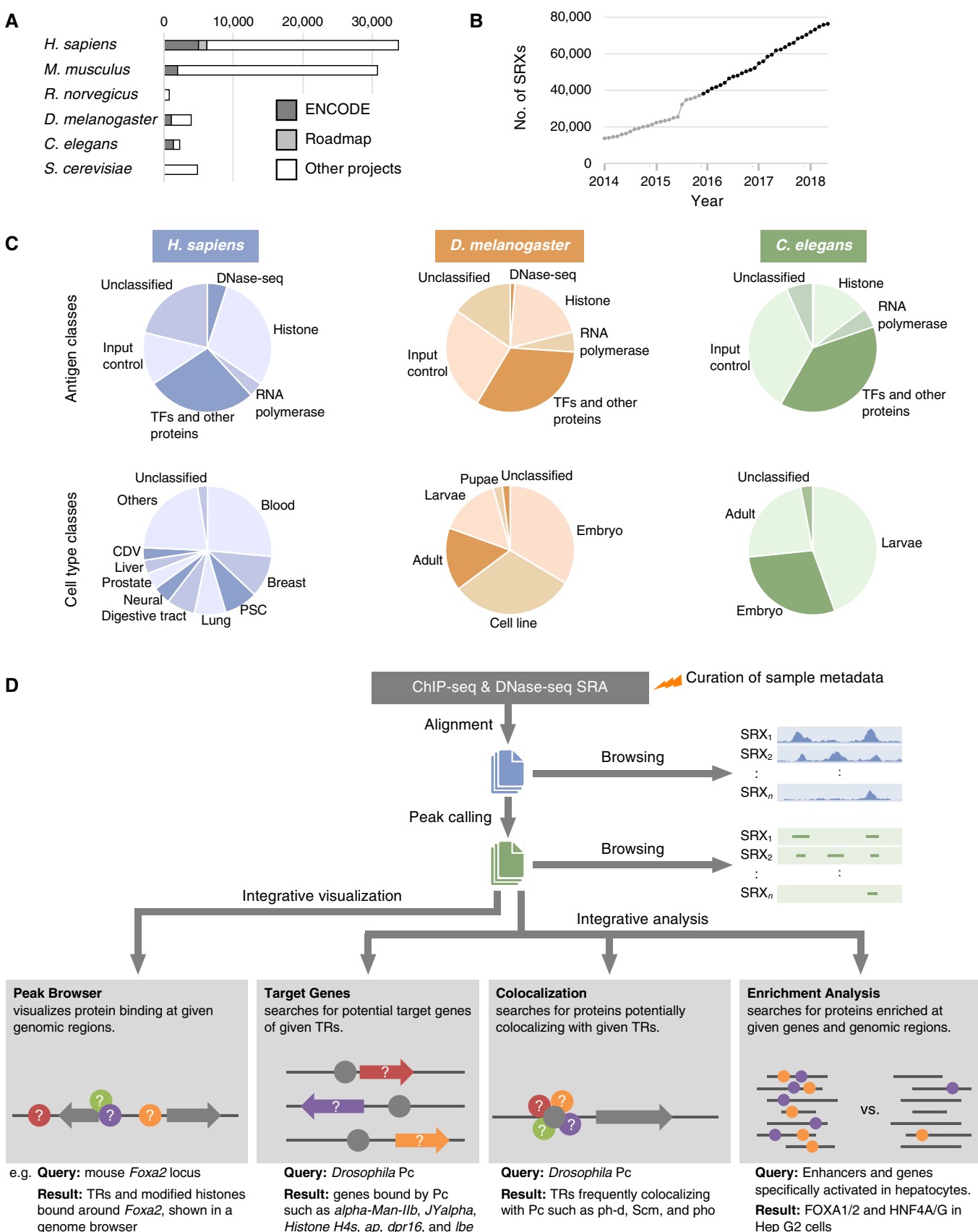

**Figure 1.**

**Figure 1.  Overview of the ChIP-Atlas data set and computational processing.**

A   Numbers of ChIP-seq and DNase-seq experiments recorded in ChIP-Atlas (as of May 2018), indicating the proportion of the data for each species derived from ENCODE, Roadmap Epigenomics, and other projects.

B   Cumulative number of SRX-based experiments recorded in ChIP-Atlas. Data published before and after the launch of ChIP-Atlas in December 2015 are shown in gray and black, respectively.

C   Numbers of experiments according to antigen (top) or cell type (bottom) classes for human, fruit fly, and nematode data. PSC, pluripotent stem cell; CDV, cardiovascular.

D   Overview of data processing. Raw sequence data are downloaded from NCBI SRA, aligned to a reference genome, and subjected to peak calling, all of which can be monitored with the genome browser IGV. All peak-call data are then integrated for browsing via the "Peak Browser" function, and they can be analyzed for TR–gene ("Target Genes") or TR–TR ("Colocalization") interactions as well as subjected to enrichment analysis ("Enrichment Analysis"). All of the results are tagged with curated sample metadata such as antigen and cell type names. In the diagrams, gray components (circles, TRs; arrows, genes) indicate queries by the user, with colored components representing the returned results.

## Results and Discussion

### Overview of the data set and design of analyses

SRA of NCBI is a huge data resource that collects all public raw sequencing data from high-throughput sequencing experiments including ChIP-seq and DNase-seq. It thus covers all sequence data presented in academic papers; data produced by large consortia (such as ENCODE and Roadmap Epigenomics [4–7,12]) and some other grant-in-aid projects; and data deposited in NCBI GEO (https://www.ncbi.nlm.nih.gov/gds), EBI ArrayExpress (https://www.ebi.ac.uk/arrayexpress), and DDBJ (https://www.ddbj.nig.ac.jp). ChIP-Atlas collects all ChIP-seq and DNase-seq data archived in NCBI SRA, with samples derived not only from human and mouse, which are also covered in other existing databases (ReMap, Cistrome DB, and GTRD [8,9,11]), but also from four other model organisms (rat, fruit fly, nematode, and budding yeast; Fig 1A and

Table 1). In NCBI SRA, each experiment is assigned an ID with a prefix of SRX, DRX, or ERX (hereafter collectively referred to as SRX), which are also adopted in ChIP-Atlas for unified management of the records. The number of SRXs collected in ChIP-Atlas is 76,217 for the six organisms, which corresponds to 89.9% of the total number of ChIP-seq and DNase-seq SRXs in NCBI SRA for all organisms ($n = 84,826$ as of May 2018). Since the public release of ChIP-Atlas, the data have been updated monthly concurrent with the monthly update of NCBI SRA (Fig 1B). We manually curate the names of antigens and cell types according to commonly or officially adopted nomenclature. The antigens and cell types are further sorted into "antigen classes" and "cell type classes", allowing categorization and extraction of data for given classes (Figs 1C and EV1B). To complete the monthly curation in an expeditious and precise manner, we developed a database and conversion tool that are specialized to return controlled vocabularies from given synonyms of TRs and cell lines or other keywords (such as catalog

**Table 1.   Comparison of ChIP-Atlas with other ChIP-seq databases.**

| | ChIP-Atlas | Cistrome DB | ReMap | GTRD |
|---|---|---|---|---|
| Data source | NCBI SRA fully encompassing GEO, ArrayExpress, DDBJ, ENCODE, and Roadmap Epigenomics data | GEO, ENCODE, and Roadmap Epigenomics | GEO, ArrayExpress, and ENCODE | GEO, ENCODE, and a part of SRA |
| Experiments | ChIP-seq and DNase-seq | ChIP-seq, DNase-seq, and ATAC-seq | ChIP-seq for TRs | ChIP-seq for TRs |
| Filtering of data for quality control | No | Yes | Yes | No |
| Number of experiments | 76,217 | 20,535 | 3,180 | 12,168 |
| Organism[a] | Hs, Mm, Rn, Dm, Ce, and Sc | Hs and Mm | Hs | Hs and Mm |
| Genome assembly | hg19, mm9, rn6, dm3, ce10, and sacCer3 | hg38 and mm10 | hg38 and hg19 | hg38 and mm10 |
| Peak caller | MACS2 | MACS2 | MACS2 | MACS, SISSRs, GEM, and PICS |
| Display format for each experiment | Alignment and peaks | Alignment and peaks | Peaks | Peaks |
| Browsing assembled peaks | Possible | None | Possible | Possible |
| Genome browser | IGV and UCSC[b] | UCSC | UCSC, ENSEMBL, and IGV | Self-developed |
| Integrative analysis tools | Search tool for target genes and colocalizing factors of given TR, and enrichment analysis tool for given genes and genomic coordinates | Search tool for target genes of given single experiment | Enrichment analysis tool for given genes and genomic coordinates relative to random background | Search tool for target genes of given TR |

[a]Hs, *Homo sapiens*; Mm, *Mus musclus*; Rn, *Rattus norvegicus*; Dm, *Drosophila melanogaster*; Ce, *Caenorhabditis elegans*; Sc, *Saccharomyces cerevisiae*.
[b]Track hub URL = http://fantom.gsc.riken.jp/5prim/external/ChIP-Atlas/current/hub.txt.

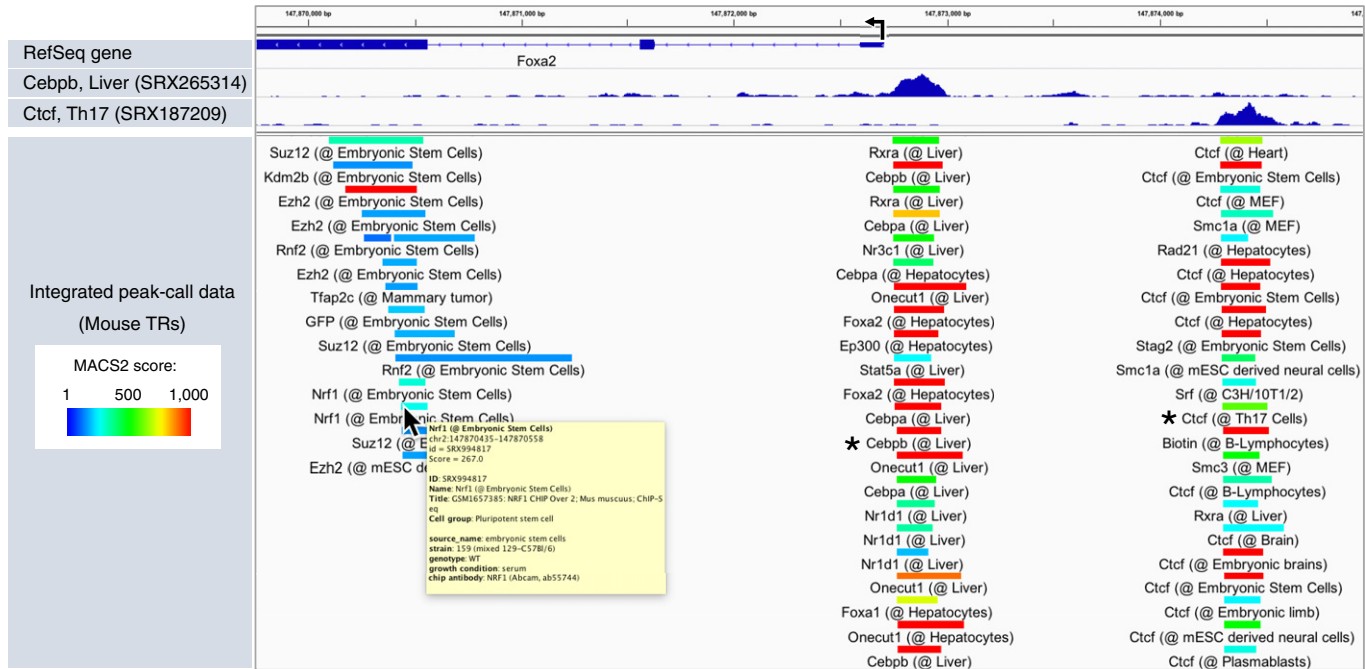

**Figure 2.  Example of processed data visualized with "Peak Browser" of ChIP-Atlas.**

ChIP-Atlas peak-call data for TRs around the mouse *Foxa2* locus are shown in the IGV genome browser for settings of the "Peak Browser" Web page shown in Fig EV1C. Bars represent the peak regions, with the curated names of the antigens and cell types being shown below the bars and their color indicating the score calculated with the peak-caller MACS2 ($-\log_{10}[Q$-value]). Detailed sample information (yellow window) appears on placing the cursor over each bar. Clicking on the bars (asterisks) enables display of the alignment data (top) and detailed information about the experiments (Fig EV1D).

numbers of antibodies and abbreviations of cell or tissue names) described in SRA sample metadata by original data submitters. The sequence data are aligned to a reference genome with Bowtie2 [13] and subjected to peak calling with MACS2 [14], and the results are readily downloaded and browsed in the genome browser IGV [15] (Figs 1D and 2 top) by entering the SRX ID or a given keyword (or keywords) in the corresponding search page of ChIP-Atlas (Fig EV1A, B and D).

Furthermore, notable features of ChIP-Atlas are that it allows browsing of peak-call data of the entire data set with IGV as well as integrative analysis not only to reveal TR–gene and TR–TR interactions but also to allow enrichment analysis for given genomic intervals based on global protein–genome binding data (the four functions shown in Fig 1D), as is described below with some examples.

**Visualization of assembled peak-call data**

All peak-call data recorded in ChIP-Atlas can be graphically displayed with the "Peak Browser" function at any genomic regions of interest (ROIs). To implement this function, we integrated a large amount of peak-call data (499, 334, 1.27, 1.39, 3.87, and 0.59 million peaks for the human, mouse, rat, fruit fly, nematode, and budding yeast genomes, respectively), indexed them for IGV, and constructed a Web interface that externally controls IGV preinstalled on the user's machine (Mac, Windows, or Linux platforms). For instance, on specification of ChIP-seq data for mouse TRs on the Web page (Fig EV1C), the corresponding results are streamed into

IGV as shown in Fig 2, suggesting that the mouse *Foxa2* gene promoter is bound by multiple TRs in the liver (Fig 2, center), that expression of the gene is suppressed by Polycomb group 2 proteins such as Suz12 and Ezh2 in embryonic stem cells (Fig 2, left), and that the upstream region of *Foxa2* may possess insulator activity due to Ctcf binding in multiple cell types (Fig 2, right). The colors of peaks indicate the statistical significance values calculated by the peak-caller MACS2 (MACS2 scores), and the names of antigen and cell types are clearly shown beneath the peaks. Clicking on a peak opens a Web page containing detailed information including sample metadata, library description, and read quality (Fig EV1D) as well as controllers to display the alignment data in IGV (Fig 2, top). Assembled peak-call data can also be browsed via the "My hubs" function of the UCSC Genome Browser (http://genome-asia.ucsc.edu/cgi-bin/hgHubConnect) by entering a URL for the ChIP-Atlas track hub (http://fantom.gsc.riken.jp/5prim/external/ChIP-Atlas/current/hub.txt) [16,17]. ChIP-Atlas thus allows not only visualization of the data for each experiment but also browsing of an integrative landscape of multiple chromatin-profiling results, potentially providing insight into the location of functional regions (enhancers, promoters, and insulators) and the corresponding regulatory factors (TRs and histone modifications).

**TR–gene and TR–TR interactions**

The large number of peak sets is further subjected to integrative analyses for data mining (Fig 1D). All TR peaks are examined for whether they are located around (±1, 5, or 10 kb) transcription start

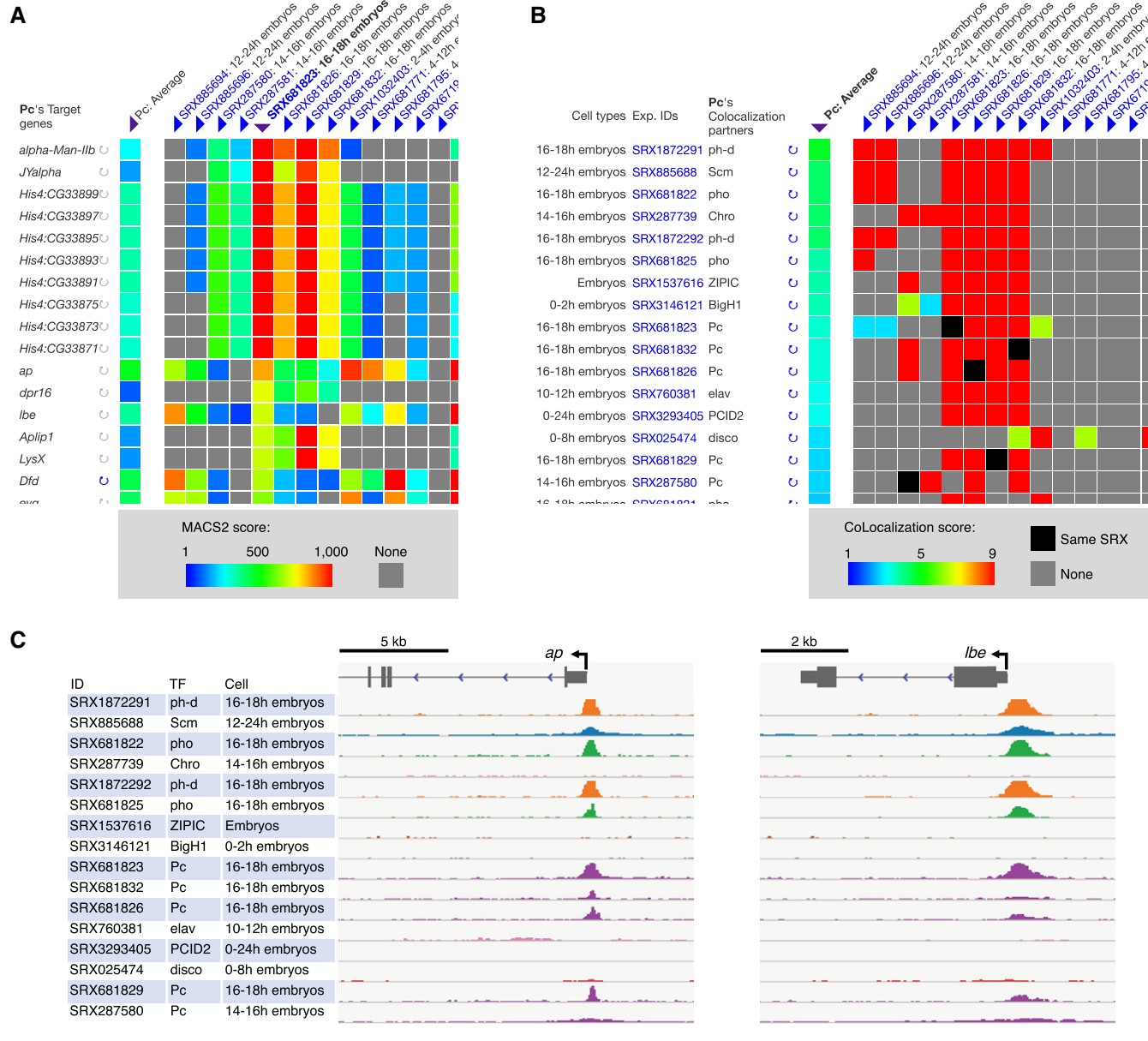

**Figure 3. Examples of analysis with "Target Genes" and "Colocalization" of ChIP-Atlas.**

A   Potential target genes of *Drosophila* Pc are listed on the left with ChIP-seq data. The colors of the cells of the matrix indicate the MACS2 scores for Pc ChIP-seq peaks (columns) within TSS ± 1 kb regions of each potential target gene (rows). As the default, the matrix is sorted according to the average of MACS2 scores in each row ("Pc: Average" at top left). Resorting is also possible by clicking the triangles under the SRX of interest at the top (sorted result for SRX681823 is shown). This table was obtained with the queries shown in Fig EV2A.

B   TRs that potentially colocalize with *Drosophila* Pc are listed on the left with their ChIP-seq information. Each cell of the matrix indicates the similarity between the ChIP-seq data for Pc (columns) and those for its potential colocalizing partners (rows) as shown by heat colors and calculated with CoLo. As the default, the matrix is sorted according to the average of CoLo scores in each row ("Pc: Average" at top left as shown here). Sorting by an SRX of interest is possible by clicking the triangles at the top. This table was obtained with the queries shown in Fig EV2B.

C   IGV snapshots showing the alignment data (BigWig format) around the *Drosophila ap* and *lbe* gene loci for ChIP-seq experiments listed on the left in (B). The results suggest that both genes might be regulated by Pc together with its colocalization partners (ph-d, Scm, and pho). The y-axes range from 0–10 RPM units.

sites (TSSs) of RefSeq coding genes, with the summarized results being provided by the "Target Genes" function of ChIP-Atlas. For example, on selection of *Drosophila* Pc (also known as Polycomb) as a query TR, and TSS ± 1 kb as the target range (Fig EV2A), this service displays genes with TSS ± 1 kb regions bound by Pc. As the

default, the potential target genes are sorted by MACS2 score averaged over all the Pc ChIP-seq data ($n = 36$; shown in the "Pc: Average" column of Fig 3A). The results can be resorted for an SRX of interest. For example, selection of SRX681823 (ChIP-seq data for Pc in 16- to 18-h embryos) (Fig 3A) resorts potential target genes such

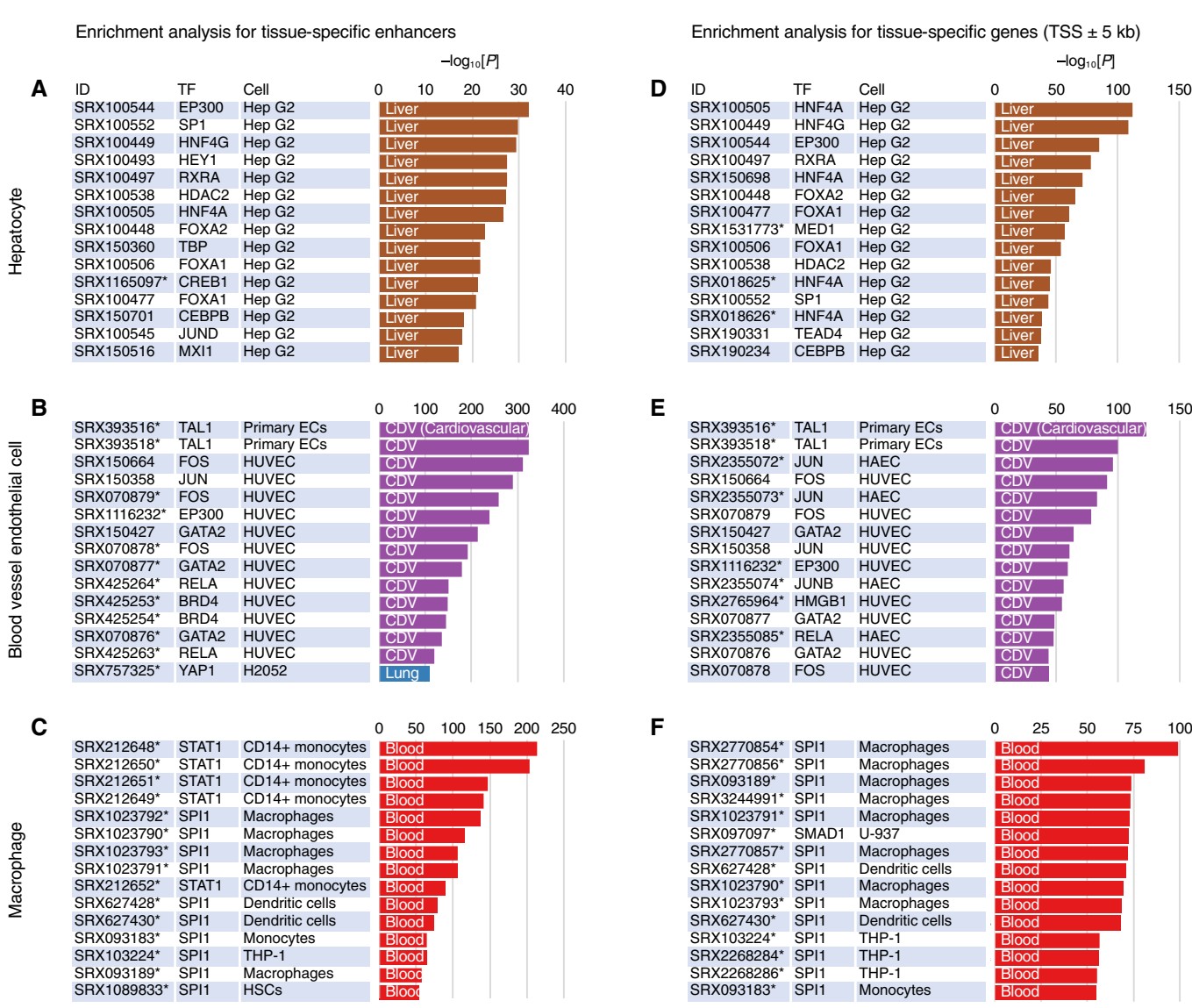

**Figure 4.  Analysis of TR enrichment at tissue-specific enhancers and genes with "Enrichment Analysis" of ChIP-Atlas.**

A–F   The top 15 ChIP-seq experiments enriched for enhancers (A–C) or genes (D–F) specifically activated in hepatocytes (A and D), blood vessel endothelial cells (B and E), or macrophages (C and F) relative to all other FANTOM5 enhancers (A–C) or RefSeq coding genes (D–F) are shown. The bar charts indicate *P*-values for enrichment, with the colors indicating the cell types examined in the experiments according to the palette shown in Fig EV4, where the top 50 ChIP-seq experiments enriched for the above and other enhancers are also presented. Asterisks next to SRX IDs indicate that the ChIP-seq data originated from projects other than ENCODE or Roadmap Epigenomics.

as *alpha-Man-IIb*, *JYalpha*, genes encoding Histone H4s, *ap*, *dpr16*, and *lbe* in order of MACS2 score. Of note, multiple ChIP-seq data can be compared in a single view as shown in Fig 3A, where *ap* and *lbe* loci both appear to be bound by Pc at various stages of embryonic development. It should be noted, however, that the genes listed by "Target Genes" are not necessarily functional targets of a given TR and that actual regulation of potential target genes would need to be confirmed experimentally such as by analysis of cells deficient in the TR.

Integrative analysis is also applied to search for sets of TRs that potentially colocalize in a genome-wide manner. Pairwise comparisons of all SRXs are thus performed with the CoLo algorithm

(https://github.com/RyoNakaki/CoLo; R. Nakaki, in preparation), and the similarity scores are precomputed for all combinations. For instance, selection of *Drosophila* Pc protein in embryos as a query on the "Colocalization" Web page of ChIP-Atlas (Fig EV2B) reveals that the ChIP-seq data are similar to those for other Polycomb group proteins such as ph-d, Scm, and pho (Fig 3B). These TRs are shown to be colocalized with Pc around its target genes such as *ap* and *lbe* gene loci (Fig 3C), suggestive of cooperative repression of these genes at embryonic stages. This function is thus useful to examine TR–TR interactions and genome-wide colocalization among public ChIP-seq data. In addition, this function can compare multiple ChIP-seq data for the same TR in a single view, which is helpful to assess

read quality, the applied antibodies, and other experimental conditions associated with similar or different binding profiles.

## Enrichment analysis for given loci and genes

"Enrichment Analysis" of ChIP-Atlas is a tool that allows a search for histone modifications and TRs enriched at a batch of genomic ROIs. On submission of two sets of genomic regions (ROIs and background regions), this service evaluates all SRXs to count the overlaps between the peaks and submitted regions, before returning enrichment analysis data including SRX IDs, antigens, cell types, and *P*-values (Fig EV2C–E). As a proof of principle, we selected human hepatocyte-specific enhancers as the ROIs ($n = 286$) and those activated in other tissues as the background ($n = 20,509$; these enhancers were obtained from FANTOM5 "predefined enhancer data" [18]), and we applied these selections to the Enrichment Analysis. Significantly enriched TRs included HNF4A/G and FOXA1/2 ($P < 1 \times 10^{-21}$; Figs 4A and EV3A), which are required for liver development and are able to directly reprogram skin fibroblasts into hepatocyte-like cells [19–21]. Furthermore, TRs for the top 15 ranked SRXs included SP1, RXRA, CEBPB, and JUND, all of which function in the liver–biliary system according to Mouse Genome Informatics (MGI) phenotype collections (MP:0005370). In addition, the predominant cell type from which enriched SRXs were derived was Hep G2 in the Liver class, even though the number of experiments in this class is relatively small for human (Fig 1C). On submission of coordinates for FANTOM5 enhancers specifically activated in blood vessel endothelial cells, all cell types of the top 14 ranked SRXs were endothelial cells in the Cardiovascular class (Fig 4B), with the enriched TRs (TAL1, JUN, EP300, GATA2, and YAP1) being related to blood vessel morphology phenotypes (MP:0001614). In the case of FANTOM5 macrophage-specific enhancers, STAT1 in monocytes and SPI1 (also known as PU.1) in macrophages were significantly enriched (Fig 4C), consistent with the fact that SPI1 is able to reprogram fibroblasts into macrophages [22]. Enrichment analysis for other cell type-specific FANTOM5 enhancers is shown in Fig EV4, with enrichment of ChIP-seq data of the Blood class being apparent for enhancers of other blood cell types such as dendritic cells, monocytes, and T cells.

In addition to genomic coordinates in BED format, ChIP-Atlas Enrichment Analysis can accept gene symbols as input to search for histone modifications and TRs with enriched binding within the specified target range of TSSs of the genes (Fig EV2D). We prepared lists of genes specifically expressed in hepatocytes, endothelial cells, or macrophages according to the FANTOM5 "promoter atlas data" [23]. We submitted these lists to Enrichment Analysis (target range of TSS ± 5 kb; other RefSeq coding genes as background), and we obtained similar results as with the corresponding enhancers (Fig 4D–F). Enrichment Analysis is thus useful to identify potential factors that collectively regulate gene sets with common features and properties.

## Examples of other uses and data sharing

In addition to the above-mentioned analyses, ChIP-Atlas Enrichment Analysis (formerly designated "*in silico* ChIP") has been used for various other purposes. For example, this tool has been applied to analyze TR enrichment at genomic ROIs such as expression quantitative trait loci (eQTLs), a user's own ChIP-seq peak-call data, and

evolutionarily accelerated regions [24–26] as well as genes whose expression levels are changed by drug exposure, aging, or cancer development [27–29] (see http://chip-atlas.org/publications for the full list of publications citing ChIP-Atlas). The results generated by ChIP-Atlas are all assigned unique URLs (see ChIP-Atlas document in https://github.com/inutano/chip-atlas/wiki for details) and are publicly available, and they are thus ready for sharing seamlessly among researchers for subsequent analysis in command lines and for interconnecting with other biodatabases such as the DeepBlue Epigenomic Data Server and RegulatorTrail [30,31], where ChIP-Atlas data have been imported and subjected to analyses.

## Comparison to relevant Web services

Before and after public release of ChIP-Atlas, several similar Web services have been offered, of which Cistrome DB, ReMap, and GTRD provide thousands of analyzed ChIP-seq data sets [8,9,11] (Table 1). ChIP-Atlas covers more ChIP-seq data and organisms than these other services, but, unlike Cistrome DB, it does not cover ATAC-seq data. Whereas preprocessed peak-call data of each experiment can be shown by all four services, alignment data are available only in Cistrome DB and ChIP-Atlas. Although assembled peak-call data can be browsed with ReMap, GTRD, and ChIP-Atlas, ChIP-Atlas Peak Browser more clearly shows the names of antigen and cell types beneath the peaks (Fig 2). Integrative analysis tools are also available in all four services. Potential target genes can be analyzed by Cistrome DB, GTRD, and ChIP-Atlas (Table 1), among which the ChIP-Atlas Target Genes function has a feature that allows comparison of target genes for multiple ChIP-seq data from various cell types in a single view (Fig 3A). Enrichment analysis in a graphical user interface is possible with ChIP-Atlas as well as with LOLAweb [10] and "Annotation Tool" of ReMap. Enrichment analysis for hepatocyte enhancers by LOLAweb, for which the data source is mainly human and mouse data of ENCODE and Cistrome DB, yields results (FOXA2, SP1, and RXRA in Hep G2) similar to those obtained by ChIP-Atlas (Figs 4A and EV5A). Annotation Tool of ReMap calculates *P*-values relative to a random background. Enrichment analysis for hepatocyte enhancers by ReMap also presented the same TRs (HNF4A, SP1, HDAC2, and RXRA) as did ChIP-Atlas (Fig EV5B), but cell type information was not provided. Although ChIP-Atlas allows such random permutation analysis (see "Random permutation of user data" in Fig EV2C), this is not recommended unless the user has appropriate background data, given that most randomly selected regions are devoid of TR binding (Fig EV3B). We are currently improving ChIP-Atlas to make it compatible with the latest versions of reference genomes such as hg38 for human and mm10 for mouse, as is the case for other services. Given that we designed ChIP-Atlas to be updated monthly with semiautomatic pipelines and systematic curation, the source data will be continuously expanded and may include more sophisticated chromatin-profiling results.

# Materials and Methods

### Source data and primary processing

Sample metadata and biosample data of all SRXs were downloaded from NCBI FTP sites (ftp://ftp.ncbi.nlm.nih.gov/sra/reports/Metadata

and ftp://ftp.ncbi.nlm.nih.gov/biosample). ChIP-Atlas uses SRXs that meet the following criteria: LIBRARY_STRATEGY is "ChIP-seq" or "DNase-Hypersensitivity"; LIBRARY_SOURCE is "GENOMIC"; taxonomy_name is "*Homo sapiens*", "*Mus musculus*", "*Rattus norvegicus*", "*Caenorhabditis elegans*", "*Drosophila melanogaster*", or "*Saccharomyces cerevisiae*"; and INSTRUMENT_MODEL includes "Illumina" (the proportion of non-Illumina platforms is ~5%). Binarized sequence raw data (.sra) for each SRX were downloaded and decompressed into Fastq format with the "fastq-dump" command of SRA Toolkit (http://www.ncbi.nlm.nih.gov, ver. 2.3.2-4) according to a default mode, with the exception of paired-end reads, which were decoded with the "–split-files" option. In the case of an SRX including multiple runs, decompressed Fastq files were concatenated into a single such file. Fastq files were then aligned with the use of Bowtie2 (ver. 2.2.2) [13] according to a default mode, with the exception of paired-end reads, for which two Fastq files were specified with "−1" and "−2" options. The following genome assemblies were used for alignment and subsequent processing: hg19 (*H. sapiens*), mm9 (*M. musculus*), rn6 (*R. norvegicus*), dm3 (*D. melanogaster*), ce10 (*C. elegans*), and sacCer3 (*S. cerevisiae*). Resultant SAM-formatted files were binarized into BAM format with SAMtools (ver. 0.1.19; samtools view) [32] and sorted (samtools sort) before removal of polymerase chain reaction (PCR) duplicates (samtools rmdup). BedGraph-formatted coverage scores were calculated with BEDTools (ver. 2.17.0; genomeCoverageBed) [33] in RPM (reads per million mapped reads) units with the "-scale 1000000/N" option, where N is mapped read counts after removal of PCR duplicates. BedGraph files were binarized into BigWig format with the UCSC BedGraphToBigWig tool (http://hgdownload.cse.ucsc.edu, ver. 4). BAM files were also used for peak calling with MACS2 (ver. 2.1.0, macs2 callpeak) [14] in BED4 format. Options for *Q*-value threshold ($< 1 \times 10^{-5}$, $< 1 \times 10^{-10}$, or $< 1 \times 10^{-20}$) were applied, and the following options were set for genome size: "-g hs" (*H. sapiens*), "-g mm" (*M. musculus*), "-g 2.15e9" (*R. norvegicus*), "-g dm" (*D. melanogaster*), "-g ce" (*C. elegans*), and "-g 12100000" (*S. cerevisiae*).

### Curation of sample metadata

Sample metadata of all SRXs (biosample_set.xml) were downloaded (ftp://ftp.ncbi.nlm.nih.gov/biosample) to extract the attributes for antigens and antibodies (http://dbarchive.biosciencedbc.jp/kyushu-u/metadata/ag_attributes.txt) as well as cell types and tissues (http://dbarchive.biosciencedbc.jp/kyushu-u/metadata/ct_attributes.txt). For sorting out of metadata dependent on the discretion of the submitter, antigens and cell types were manually annotated by curators who had been fully trained in molecular and developmental biology. Each annotation has a "Class" and "Subclass" as shown in antigenList.tab and celltypeList.tab, which are available from the document page of ChIP-Atlas (https://github.com/inutano/chip-atlas/wiki). Guidelines for antigen annotation included the following: (i) Histones were based on Brno nomenclature (such as H3K4me3 and H3K27ac) [34]; (ii) gene symbols were used for gene-encoded human proteins according to HGNC (http://www.genenames.org) (for example, OCT3/4 → POU5F1, and p53 → TP53); and (iii) modifications such as phosphorylation were ignored. If an antibody recognizes multiple molecules in a family, the first in an ascending order was chosen (for example, anti-SMAD2/3 → SMAD2).

Most human, mouse, and rat cell types were classified according to their tissue of origin. Embryonic stem cells and induced pluripotent stem cells were classified in the "Pluripotent stem cell" class. The nomenclature of cell lines and tissue names was standardized according to the following frameworks and resources: (i) Supplementary table S2 of a paper [35] proposing unified cell line names (for example, MDA-231, MDA231, or MDAMB231 → MDA-MB-231); (ii) ATCC, a nonprofit repository of cell lines (http://www.atcc.org); (iii) cell line nomenclature used in the ENCODE project; and (iv) MESH for tissue names (http://www.ncbi.nlm.nih.gov/mesh).

Antigens or cell types were categorized as "Unclassified" if the curators could not understand attribute values, and as "No description" if there was no attribute value, with these two classes being combined and designated "Unclassified" for simplicity in the present study (Fig 1C). More details and examples are available from the document page of ChIP-Atlas (https://github.com/inutano/chip-atlas/wiki).

All source codes for ChIP-Atlas are publicly available from following link of GitHub: https://github.com/shinyaoki/chipatlas/tree/master/sh

### Peak browser

BED4-formatted peak-call data of each SRX were concatenated and converted to BED9 + GFF3 format to show MACS2 scores and sample metadata in the genome browser IGV. The descriptions are provided on our document page (https://github.com/inutano/chip-atlas/wiki).

### Target genes

BED4-formatted peak-call data of each SRX for TRs (MACS2 *Q*-value $< 1 \times 10^{-5}$) were used for this function. The location of TSSs and gene symbols were based on refFlat files (http://hgdownload.cse.ucsc.edu/downloads.html), with only protein-coding genes being applied to this analysis. The BEDTools (ver. 2.17.0) "window" command was used to search for target genes among peak-call data and the TSS library with a window size option ("-w") followed by "1000", "5000", or "10000". Peak-call data for the same antigens were collected, and MACS2 scores were indicated as heat-map colors on the Web browser. If a gene intersected with multiple peaks of a single SRX, the highest MACS2 score was chosen for the color indication. The "Average" column at the far left of the results table shows the mean of the MACS2 scores in each row.

### Colocalization

BED4-formatted peak-call data were analyzed for similarities to other peak-call data with CoLo, a tool for evaluation of the colocalization of TRs with multiple ChIP-seq peak-call data (R.N., in preparation; https://github.com/RyoNakaki/CoLo). The advantages of CoLo are that (i) it compensates for biases derived from different experimental conditions, and (ii) it adjusts the differences in peak number and distribution due to intrinsic characteristics of the TRs. Function (i) is programmed so that MACS2 scores in each BED4 file are fitted to a Gaussian distribution, with the BED4 files being divided into three groups: "H" (high binding, *Z*-score $> 0.5$), "M" (middle binding, $-0.5 \leq$ *Z*-score $\leq 0.5$), and "L" (low binding, *Z*-

score < −0.5). These three groups are treated as independent data for evaluation of similarity with function (ii). CoLo thus evaluates the similarity between two SRXs ($SRX_1$ and $SRX_2$) according to nine combinations (H, M, and L of $SRX_1$ versus H, M, and L of $SRX_2$). A set of nine Boolean results (similar or not) is returned to indicate the similarity of $SRX_1$ and $SRX_2$. The scores for comparison of the BED files are calculated by multiplication of the combination of H (= 3), M (= 2), or L (= 1); for example, if H of $SRX_1$ and M of $SRX_2$ are evaluated to be similar, the score is $3 \times 2 = 6$. If multiple H/M/L combinations were returned from $SRX_1$ and $SRX_2$, the highest score was adopted. The "Average" column on the far left of the results table shows the mean of the CoLo scores in each row.

### Enrichment analysis

On submission, the two sets of BED-formatted data (ROIs and background regions, see below) are sent to the NIG supercomputer server, where the overlaps between the submitted data and peak-call data archived in ChIP-Atlas are counted with the "intersect" command of BEDTools2 (ver. 2.23.0). *P*-values are calculated with the two-tailed Fisher's exact probability test (the null hypothesis is that the two data sets overlap with the ChIP-Atlas peak-call data in the same proportion). *Q*-values are calculated with the Benjamini and Hochberg method.

### Analysis of tissue-specific enhancers

FANTOM5 facet-specific predefined enhancer data were downloaded from PrESSTo (http://pressto.binf.ku.dk) and submitted to the Enrichment Analysis Web page with the following parameter settings: "Antigen Class" is "TFs and others"; "Cell type Class" is "All cell types"; and "Threshold for Significance" (MACS2 score) is "100". The background data for comparison were prepared by assembling FANTOM5 enhancers of all facets with the exception of the target facet.

### Analysis of tissue-specific genes

FANTOM5 CAGE data (http://fantom.gsc.riken.jp/5/datafiles/phase 1.3/extra/Sample_ontology_enrichment_of_CAGE_peaks) were downloaded, and the locations of CAGE peaks were converted to the corresponding official gene symbols with the library files (http://fantom.gsc.riken.jp/5/datafiles/phase1.3/extra/TSS_classif ier). They were then submitted to the Enrichment Analysis Web page with the following parameter settings: "Antigen Class" is "TFs and others"; "Cell type Class" is "All cell types"; "Threshold for Significance" (MACS2 score) is "100"; the background data for comparison are "Refseq coding genes (excluding user data)"; and "Distance range from TSS" is "TSS ± 5,000 bp".

**Expanded View** for this article is available online.

### Acknowledgements

The public data used in this research were obtained from SRA of NCBI. Computations were performed mostly on the NIG supercomputer at ROIS National Institute of Genetics. This work was supported by KAKENHI grants 25840087, 18KT0024, 16H06530, 26291051, 17H01571, and 15K14529 from the Japan Society for the Promotion of Science (JSPS); by AMED under grant number JP17gm5010003; and by the National Bioscience Database Center (NBDC) of the Japan Science and Technology Agency (JST).

### Author contributions

SO, TO, and GS conceived and designed the database. SO, TO, and CM annotated the sample metadata. SO, TO, HH, OO, YO, HK, and RN implemented the database and data transport pipeline. SO, TO, GS, JS, and CM performed the analyses. SO and CM wrote the manuscript.

### Conflict of interest

The authors declare that they have no conflict of interest.

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
