## [Review Process File · EMBO Reports]

ChIP-Atlas: a data-mining suite powered by full integration of public ChIP-seq data

Shinya Oki, Tazro Ohta, Go Shioi, Hideki Hatanaka, Osamu Ogasawara, Yoshihiro Okuda, Hideya Kawaji, Ryo Nakaki, Jun Sese & Chikara Meno

Review timeline:

Submission date:	9 April 2018
Editorial Decision:	7 June 2018
Revision received:	9 August 2018
Editorial Decision:	14 September 2018
Revision received:	3 October 2018
Accepted:	12 October 2018

Editor: Esther Schnapp

Transaction Report:

1st Editorial Decision

7 June 2018

Thank you for the transfer of your manuscript to EMBO reports. We have now received the enclosed comments from the referees.

As you will see, both referees acknowledge that ChIP-Atlas is a valuable resource for the community. However, they also point out that the advantage and strength of ChIP-Atlas should be worked out better, especially in comparison with existing, similar databases.

We would thus like to invite you to revise your manuscript with the understanding that the referee concerns must be fully addressed and their suggestions taken on board. Please address all referee concerns in a complete point-by-point response. Acceptance of the manuscript will depend on a positive outcome of a second round of review. It is EMBO reports policy to allow a single round of revision only and acceptance or rejection of the manuscript will therefore depend on the completeness of your responses included in the next, final version of the manuscript.

Revised manuscripts should be submitted within three months of a request for revision; they will otherwise be treated as new submissions. Please contact us if a 3-months time frame is not sufficient for the revisions so that we can discuss this further. You can either publish the study as a short report or as a full article. For short reports, the revised manuscript should not exceed 27,000 characters (including spaces but excluding materials & methods and references) and 5 main plus 5 expanded view figures. The results and discussion sections must further be combined, which will help to shorten the manuscript text by eliminating some redundancy that is inevitable when discussing the same experiments twice. For a normal article there are no length limitations, but it should have more than 5 main figures and the results and discussion sections must be separate. In both cases, the entire materials and methods should be included in the main manuscript file.

Please note that the EMBO reports reference style is numbered. It can be found in EndNote.

Supplementary figures, tables and movies can be provided as Expanded View (EV) files, and we can offer a maximum of 5 EV figures per manuscript. EV figures are embedded in the main manuscript

text and expand when clicked in the html version. Additional supplementary figures will need to be included in an Appendix file. Tables can either be provided as regular tables, as EV tables or as Datasets. Please see our guide to authors for more information.

- a complete author checklist, which you can download from our author guidelines (<http://embor.embopress.org/authorguide#revision>). Please insert page numbers in the checklist to indicate where in the manuscript the requested information can be found. The completed author checklist will also be part of the RPF (see below).
- a letter detailing your responses to the referee comments in Word format (.doc)
- a Microsoft Word file (.doc) of the revised manuscript text
- editable TIFF or EPS-formatted figure files in high resolution. In order to avoid delays later in the process, please read our figure guidelines before preparing your manuscript figures at: http://www.embopress.org/sites/default/files/EMBOPress_Figure_Guidelines_061115.pdf

I look forward to seeing a revised version of your manuscript when it is ready. Please let me know if you have questions or comments regarding the revision.

REFeree REPORTS

Referee #1:

The authors present a great resource of ChIP-Seq data in 6 model organisms. While such a resource exists for human data, it is very unique to have the TFs for so many different species in one place. The manuscript itself focusses mainly on human and mouse TFs and presents several case studies to illustrate the power of the database. Overall, this will be (or is already) a very valuable resource for the community.

One of the strong points of the ChIP-Atlas presented here is the resource for non-human TFs.

- Much of the intro/abstract is focused on "human genome" whereas the strengths of the study is the collection of non-human TF experiments.
- For the human TFs there is already a well run database (REMAP) that should also be mentioned by the authors and compared to their database, the newest update (2017) includes ENCODE and non-ENCODE human TFs and should be cited
- "in silico ChIP" is a strange term for saying "looking for enrichment of TF binding using publicly available ChIP-Seq data"
- "unique tool to search for TFs enriched at a batch of genomic ROIs." is quite an overstatement. There are several tools that do similar things, and it is not difficult to calculate in 2 lines of code if the data sets are available. It is "useful" tool, just not "unique"
- the authors claim that detecting enriched cell types, for which only a small set of experiments has

been performed illustrates the power of the method. I would rather want to see whether the method is also able to identify the correct cell type if many experiments have been performed (which makes it inherently more noisy).

- how are the background regions defined for the enrichments? Did the authors perform any benchmarking for different types of background definitions?

- what are the quality controls for the data sets that were included in the study? A workflow figure and description of the QCs for this would be useful in the main text

- given the many different species that are represented in the resource it could be interesting to add some tools about conservation to the website - not for the current manuscript but for a future outlook of the site.

Referee #2:

As pointed out by the other reviewers, I strongly believe that the database developed by the authors will be of interest to the community. Nevertheless, the novelty of the presented results is very limited to say the least and other resources provide similar data. The authors tried to highlight the tools provided through the ChIP-atlas as novelty but fail short on the description and assessment of two of them (namely the target and colocalization predictions). This manuscript clearly belongs to a resource description category but do not provide any biological results.

Moreover, I have the following specific comments:

1. The first sentence of the Introduction: "The vast majority of noncoding regions in the human genome is thought to possess biological activity" is clearly an overstatement. It is particularly vexing to read it when the authors wrote in the responses to reviewers that they have deleted this sentence!

2. There is a big issue with the semantic used by the authors all along the manuscript regarding transcription factors (TFs). Indeed, the authors state that they collected data for 743 TFs and always refer to the TFs in their analyses. But part of their data is based on proteins that are **not** TFs (e.g. SMC3, SMC4, RAD21, EP300). The authors should refer to the latest curation of TFs provided by Lambert et al., Cell, 2018.

Moreover, the abstract reads that they provide 97 million binding sites while they provide binding regions and **not** binding sites. Indeed, TF binding sites are the actual (short) DNA sequences specifically bound by the TFs.

3. I am really surprised that the authors did not provide a clear description of the state of the art regarding other very similar databases storing uniformly processed ChIP-seq (and others). The authors should clearly put ChIP-atlas into the context of other available resources such as ReMap, GTRD, CistromeDB, etc.

4. Other resources such as ReMap, GTRD, and CistromeDB uniformly processed their data using the most up-to-date reference genomes (e.g. hg38). Could the authors argument why they used hg19 for instance since hg38 has been available from Dec. 2013? Further hg38 strongly improves read mapping quality.

5. As pointed out by a previous reviewer, the name "in silico ChIP" should be changed as it implies in silico simulation. It is **not** because this name has been used in other manuscript that it makes it valid. Moreover, this tool should be assessed against already existing tools doing similar enrichment analyses and already published.

6. While the cherry-picked case studies highlighted in the manuscript provide evidence that the data can be useful, it does not highlight new biology. Moreover, the long list of specific examples make the manuscript hard to read. Focusing more deeply on the data in the database would be more relevant.
7. The authors stated in their response to reviewer #2 comment #5 that in contrast to other studies, they relied on *real* TF binding data. How come they write that their data are more *real* than other experimental data processed in other studies?
8. The manuscript highlights binding "hotspots" of potential interest for GWAS analyses. As their data are specifically ChIP-seq data, the authors should at least discuss the fact that "hotspot" regions (aka HOT regions) and ChIP-seq peaks in general can be derived from ChIP-seq artifacts (see for instance Teytelman et al, PNAS; Jain et al., NAR; Hunt and Wasserman, Genome Biology; Wreczycka et al., bioRxiv; Park et al., PLOS one).
9. A large portion of the discussion actually described results and do not correspond to a proper discussion of the work.
10. Can the authors describe how they are going to monthly curate the metadata for updating ChIP-atlas?
11. In the Introduction, it reads that "integrative analysis of such data requires *sophisticated skills*. I am really skeptical about the fact that it is really *sophisticated*."
12. How come the analyses of ChIP-seq and DNase-seq data for six organisms provided in ChIP-atlas correspond to *90*% of all experiments in these organisms?

1st Revision - authors' response

9 August 2018

We thank the reviewers for their constructive comments. The main criticisms of our manuscript seem to be that (1) it would be more suitable as a resource paper rather than a research paper providing novel biological insights, and (2) comparison of ChIP-Atlas with other relevant tools was lacking. With the permission of the editor, we have therefore decided to submit the manuscript as a "resource paper" in the Scientific Report format of *EMBO Reports*. The major points of revision in the manuscript are as follows:

- The title and abstract have been changed to be more suitable for a resource paper.
- Functions of ChIP-Atlas that we omitted in the former manuscript ("Target Genes," "Colocalization," and "Enrichment Analysis" for a given gene set) are now described (**Figs 3 and 4E–H**).
- Analyses of data from species other than human and mouse are now shown as examples (**Figs 1C and 3**).
- Given a reviewer's comment that the section on enrichment analysis was too long, we have omitted analysis of GWAS SNPs and replaced this section with a short paragraph to show that this tool can also handle specific gene sets (**Fig 4E–H**).
- ChIP-Atlas is now compared with other relevant tools in **Table 1** as well as in the main text.
- The Results and Discussion sections have been combined to conform to the Scientific Report format. We now mainly discuss the data content and functions of ChIP-Atlas in comparison with other tools rather than biological insights gained with our platform.

Response to Referee #1

The authors present a great resource of ChIP-Seq data in 6 model organisms. While such a resource exists for human data, it is very unique to have the TFs for so many different species in one place. The manuscript itself focusses mainly on human and mouse TFs and presents several case studies to illustrate the power of the database. Overall, this will be (or is already) a very valuable resource for the community.

—We thank the reviewer for these positive comments on ChIP-Atlas.

*One of the strong points of the ChIP-Atlas presented here is the resource for non-human TFs.
- Much of the intro/abstract is focused on "human genome" whereas the strengths of the study is the collection of non-human TF experiments.*

—We now emphasize that ChIP-Atlas covers other species in addition to human and mouse. The model organisms included in ChIP-Atlas are now mentioned in the Abstract, in the main text (page 4, lines 5–9), and in the **new Table 1** in comparison with other existing tools. Furthermore, we now show classification of the data content for *D. melanogaster* and *C. elegans* in the **new Fig 1C** as well as present the results of analyses of target genes and colocalization partners for *Drosophila* Pc protein as examples (**new Fig 3**).

- For the human TFs there is already a well run database (REMAP) that should also be mentioned by the authors and compared to their database, the newest update (2017) includes ENCODE and non-ENCODE human TFs and should be cited

—As suggested by the reviewer, we now compare ChIP-Atlas with other databases and tools (Cistrome DB, ReMap, and GTRD). As shown in the **new Table 1**, key features of ChIP-Atlas are that it fully encompasses data archived in NCBI SRA, including GEO, ArrayExpress, DDBJ, ENCODE, Roadmap Epigenomics, and other published data; it covers six model organisms; and it has functions for integrative analyses based on thousands of ChIP-seq experiments. These advantages of ChIP-Atlas are also addressed in the Abstract and Results and Discussion sections (page 9).

- "in silico ChIP" is a strange term for saying "looking for enrichment of TF binding using publicly available ChIP-Seq data"

—We have now renamed this tool “Enrichment Analysis.”

- "unique tool to search for TFs enriched at a batch of genomic ROIs." is quite an overstatement. There are several tools that do similar things, and it is not difficult to calculate in 2 lines of code if the data sets are available. It is "useful" tool, just not "unique"

— We had overlooked other tools for enrichment analysis. We found that enrichment analysis can be performed in a graphical user interface with LOLAweb and “Annotation tool” of ReMap as well as with ChIP-Atlas. The results obtained with these other two tools are now compared with those of ChIP-Atlas (**Fig 4A** and **new Fig EV5**), and they reveal that LOLAweb is largely dependent on Cistrome DB (derived from ENCODE data) and that ReMap does not provide cell type information (page 9, lines 15–26).

- the authors claim that detecting enriched cell types, for which only a small set of experiments has been performed illustrates the power of the method. I would rather want to see whether the method is also able to identify the correct cell type if many experiments have been performed (which makes it inherently more noisy).

—To address this comment, we performed enrichment analysis for all facets of FANTOM5 enhancers and now present the cell type classes for the top 50 experiments of each facet (**new Fig EV4**) in addition to the top 15 (**Fig 4A–D**). Overall, enhancers specifically activated in blood, hepatic, or endothelial lineages were enriched with ChIP-seq data of Blood, Liver, and Cardiovascular classes, respectively, but other enhancers were enriched with ChIP-seq data of multiple cell types, suggesting that the enriched TRs may also regulate the analyzed enhancers in the

cell types that yielded the ChIP-seq data. We have now addressed this issue in the text of the Results and Discussion section (page 8, lines 3–8) corresponding to **Figs 4** and **EV4**.

- how are the background regions defined for the enrichments? Did the authors perform any benchmarking for different types of background definitions?

—The background regions for enrichment analysis were defined as enhancers or genes activated in tissues other than those in which the ROIs are active, as is described in **Fig EV2E**, Results and Discussion (pages 6–7), and Materials and Methods (page 13, line 13–31). In the revised manuscript, for comparison, we have added the results for enrichment analysis in which randomly permuted regions served as the background for hepatocyte-specific enhancers (**new Fig EV3B**). Most of the random background regions tended to be devoid of protein binding, as is now described in the main text (page 9, lines 23–26).

- what are the quality controls for the data sets that were included in the study? A workflow figure and description of the QCs for this would be useful in the main text

—Although read quality data are beneficial for estimation of the quality of an experiment, there is no generally accepted criterion for limits of read quality. Some ChIP-seq experiments may have been performed with low-quality antibodies and yet show high read quality. We thus intentionally did not set read quality-based limitations for archiving public sequence data. Instead, we present read quality for each experiment (**Fig EV1D**). Furthermore, “Target Genes” and “Colocalization” functions can show similarities and differences among independent ChIP-seq data for the same TR, providing an opportunity to assess the read quality, used antibodies, and other experimental conditions (see **new Fig 3** and corresponding main text [page 6, lines 21–24]).

- given the many different species that are represented in the resource it could be interesting to add some tools about conservation to the website - not for the current manuscript but for a future outlook of the site.

—The IGV genome browser has such a function to load conservation tracks. For example, the IGV menu “Load from Server > Comparative Genomics > PhastCons (Vertebrate 46 way)” is able to show the extent of conservation among 46 vertebrates for hg19. Given that this is an inherent function of IGV, we have not described it in the present manuscript, but we will introduce such usage through our Web site and communications to users. In the revised manuscript, we now mention that ChIP-Atlas data can also be browsed with the UCSC browser (page 5, lines 20–23), which likewise displays such conservation tracks.

Response to Referee #2

As pointed out by the other reviewers, I strongly believe that the database developed by the authors will be of interest to the community. Nevertheless, the novelty of the presented results is very limited to say the least and other resources provide similar data. The authors tried to highlight the tools provided through the ChIP-atlas as novelty but fail short on the description and assessment of two of them (namely the target and colocalization predictions). This manuscript clearly belongs to a resource description category but do not provide any biological results.

—According to the reviewer’s comment and the editor’s suggestion, we have submitted the revised manuscript as a “resource paper” in the Scientific Report format. In this revision, we have described all functions of ChIP-Atlas by adding “Target Genes” and “Colocalization” as well as “Enrichment Analysis” for a given gene set, which were not mentioned in the former manuscript. In addition, we deleted the section on enrichment analysis for GWAS SNPs that was found not to provide novel insight.

Moreover, I have the following specific comments:

1. The first sentence of the Introduction: "The vast majority of noncoding regions in the human genome is thought to possess biological activity" is clearly an overstatement. It is particularly

vexing to read it when the authors wrote in the responses to reviewers that they have deleted this sentence!

—We apologize for this oversight. We have now revised the Introduction to focus on the current state of public ChIP-seq data and their application in the research community.

*2. There is a big issue with the semantic used by the authors all along the manuscript regarding transcription factors (TFs). Indeed, the authors state that they collected data for 743 TFs and always refer to the TFs in their analyses. But part of their data is based on proteins that are *not* TFs (e.g. SMC3, SMC4, RAD21, EP300). The authors should refer to the latest curation of TFs provided by Lambert et al., Cell, 2018.*

*Moreover, the abstract reads that they provide 97 million binding sites while they provide binding regions and *not* binding sites. Indeed, TF binding sites are the actual (short) DNA sequences specifically bound by the TFs.*

—We now use the term “transcriptional regulators (TRs)” to encompass transcription factors and other proteins involved in gene regulation, in accordance with the ReMap paper (Chèneby et al., NAR, 2018) and as described in the Introduction section (page 3, line 2–7). The sentence including “97 million binding sites” was deleted from the Abstract so as not to focus on human data.

3. I am really surprised that the authors did not provide a clear description of the state of the art regarding other very similar databases storing uniformly processed ChIP-seq (and others). The authors should clearly put ChIP-atlas into the context of other available resources such as ReMap, GTRD, CistromeDB, etc.

—In the revised manuscript, we now compare ChIP-Atlas with other databases and tools (Cistrome DB, ReMap, and GTRD). As shown in the **new Table 1**, key features of ChIP-Atlas are that it fully encompasses data archived in NCBI SRA, including GEO, ArrayExpress, DDBJ, ENCODE, Roadmap Epigenomics, and other published data; it covers six model organisms; and it has functions for integrative analyses based on thousands of ChIP-seq experiments. These advantages of ChIP-Atlas are also addressed in the Abstract and Results and Discussion sections (page 9).

4. Other resources such as ReMap, GTRD, and CistromeDB uniformly processed their data using the most up-to-date reference genomes (e.g. hg38). Could the authors argue why they used hg19 for instance since hg38 has been available from Dec. 2013? Further hg38 strongly improves read mapping quality.

—As described in the Introduction (page 3, lines 23–26), the ChIP-Atlas project team was organized in 2014, when hg19 and mm9 assemblies were still more popular than the later versions. We therefore started to prepare the data in the former assemblies. Now that hg38 and mm10 have become the most popular versions, we are working to provide the data in these reference genome assemblies, as is now mentioned in the Results and Discussion section (page 9, lines 26–28). However, realignment and peak calling for >70,000 ChIP-seq experiments will be time consuming, and, given the spec of the supercomputer system we use, we expect the public release will be next year at the earliest.

*5. As pointed out by a previous reviewer, the name "in silico ChIP" should be changed as it implies in silico simulation. It is *not* because this name has been used in other manuscript that it makes it valid. Moreover, this tool should be assessed against already existing tools doing similar enrichment analyses and already published.*

—We have now renamed this tool “Enrichment Analysis.” In the revised manuscript, we compare ChIP-Atlas with other similar tools (LOLAweb and “Annotation tool” of ReMap) by performing enrichment analysis for hepatocyte enhancers. These latter two tools returned results similar to those of ChIP-Atlas (**Fig 4A** and **new Fig EV5**), although LOLAweb is largely dependent on Cistrome DB (thus, on ENCODE data) and ReMap shows no cell type information (page 9, lines 15–26).

6. While the cherry-picked case studies highlighted in the manuscript provide evidence that the data can be useful, it does not highlight new biology. Moreover, the long list of specific examples make

the manuscript hard to read. Focusing more deeply on the data in the database would be more relevant.

—According to the reviewer’s comment, we revised the manuscript as a resource paper. We simplified the results of “Enrichment Analysis” by deleting the section on GWAS analyses and restricting discussion of biological insights to a minimum. Instead, we now focus on data content and computational processing in the first subsection of Results and Discussion, and we now fully describe the functions of ChIP-Atlas including “Target Genes,” “Colocalization,” and “Enrichment Analysis” for a given gene set.

*7. The authors stated in their response to reviewer #2 comment #5 that in contrast to other studies, they relied on *real* TF binding data. How come they write that their data are more *real* than other experimental data processed in other studies?*

—In that response, we supposed motif analyses rather binding data by ChIP-seq. In the manuscript, we do not use the term “real TF binding data.”

8. The manuscript highlights binding "hotspots" of potential interest for GWAS analyses. As their data are specifically ChIP-seq data, the authors should at least discuss the fact that "hotspot" regions (aka HOT regions) and ChIP-seq peaks in general can be derived from ChIP-seq artifacts (see for instance Teytelman et al, PNAS; Jain et al., NAR; Hunt and Wasserman, Genome Biology; Wreczycka et al., bioRxiv; Park et al., PLOS one).

—As mentioned above, we have deleted the section on GWAS analyses.

9. A large portion of the discussion actually described results and do not correspond to a proper discussion of the work.

—In the revised manuscript, we combined the Results and Discussion sections to conform to the Scientific Report format.

10. Can the authors describe how they are going to monthly curate the metadata for updating ChIP-atlas?

—Since the public release in December 2015, we have monthly and manually curated the NCBI sample metadata as described in the first subsection of Results and Discussion and in Materials and Methods. We developed a private database and conversion tool specialized for performance of the curation work in a speedy and precise manner. As mentioned in Acknowledgements, the ChIP-Atlas project has been supported by the National Bioscience Database Center (NBDC) of the Japan Science and Technology Agency (JST) since last year, in particular for stable and continuous management of systematic curation and monthly updates.

*11. In the Introduction, it reads that "integrative analysis of such data requires *sophisticated skills*. I am really skeptical about the fact that it is really *sophisticated*.*

—We removed the word “sophisticated” (page 3, line 21).

*12. How come the analyses of ChIP-seq and DNase-seq data for six organisms provided in ChIP-atlas correspond to *90*% of all experiments in these organisms?*

—The number of ChIP-seq and DNase-seq SRXs collected in ChIP-Atlas is 76,217 for the six organisms, which accounts for 89.9% of those archived in NCBI SRA for all organisms ($n = 84,826$ as of May 2018). We have now clarified this point (page 4, lines 11–14).

Thank you for the submission of your revised manuscript. We have now received the enclosed reports from the referees, and I am happy to tell you that both support its publication now. Referee 2

still has a few suggestions that I would like you to incorporate before we can proceed with the official acceptance of your resource.

A few other changes/items are also needed:

- Please add a data availability paragraph to the materials and methods with the information on where the source code for ChIP-Atlas can be found.

- EMBO press papers are accompanied online by A) a short (1-2 sentences) summary of the findings and their significance, B) 2-3 bullet points highlighting key results and C) a synopsis image that is 550x200-400 pixels large (the height is variable). You can either show a model or key data in the synopsis image. Please note that text needs to be readable at the final size. Please send us this information along with the revised manuscript.

REFEREE REPORTS

Referee #1:

The authors have addressed the concerns/ comments I raised. It's a great tool and will be useful for the community.

Referee #2:

I would like to applaud the authors for the great resource they are providing. The manuscript has been very much improved as a resource paper. I only have a few comments to be addressed.

1. Some parts of the abstract need to be rewritten to be more accurate.
 - a. ChIP-atlas does not provide analyses of "significant TR-gene" interactions. The "Target" analysis is simply a distance association and no significance is assessed.
 - b. There is no support in the manuscript for claiming that ChIP-atlas outperforms other resources on "data integrity".
2. The authors describe ChIP-seq in the Introduction section but not DNase-seq. As ChIP-atlas relies on these two data types, it would be nice to introduce both.
3. The "Target Genes" function should be renamed. Indeed, this tool simply assigns peaks to genes based on windows around TSSs; no evidence for functionally targeting the genes is provided. Maybe use "Close Genes" or something alike.
4. The section "Enrichment analysis for given loci and genes" is very long and becomes a distraction for the reader. I would recommend to highlight a couple of analysis in the main text, then state that it has also been successfully applied to other data sets with details in supplementary material.
5. Figure 1 caption C  "Numbers" should be replaced by "Proportion" as the figure does not provide actual numbers here.
6. In Table 1, one line about "Quality Control" should be added. Indeed, it is an important information that is not provided by ChIP-atlas but others. It would be fair to provide this information to the reader.

Response to Referee #2

We thank the referee for again taking the time to review our paper. Our responses to the points raised are as follows:

I would like to applaud the authors for the great resource they are providing. The manuscript has been very much improved as a resource paper. I only have a few comments to be addressed.

—We thank the reviewer for the positive comment on our resource and paper.

1. *Some parts of the abstract need to be rewritten to be more accurate.*

a. *ChIP-atlas does not provide analyses of "significant TR-gene" interactions. The "Target" analysis is simply a distance association and no significance is assessed.*

b. *There is no support in the manuscript for claiming that ChIP-atlas outperforms other resources on "data integrity".*

—We have now removed the word “significant” (page 2, line 12) and have replaced “data integrity” with “data number” (page 2, line 14) in the abstract.

2. *The authors describe ChIP-seq in the Introduction section but not DNase-seq. As ChIP-atlas relies on these two data types, it would be nice to introduce both.*

—As suggested, we have now given a brief explanation of DNase-seq in the Introduction (page 3, lines 7–9).

3. *The "Target Genes" function should be renamed. Indeed, this tool simply assigns peaks to genes based on windows around TSSs; no evidence for functionally targeting the genes is provided. Maybe use "Close Genes" or something alike.*

—We agree that the genes identified by the “Target Genes” function are not necessarily regulated by the TRs of interest. However, the term “target genes” is also used in the sense of “binding targets” of a given TR. Furthermore, we hope the reviewer can appreciate that it would require substantial effort at this stage to rename the “Target Genes” tool on many Web pages and to renew our server system, as we have already done to rename “*in silico* ChIP” to “Enrichment Analysis” in the earlier revision. In response to the reviewer’s comment, we now clearly state that “Target Genes” does not necessarily identify functional targets, with such an identification requiring experimental validation (page 6, lines 12–14).

4. *The section "Enrichment analysis for given loci and genes" is very long and becomes a distraction for the reader. I would recommend to highlight a couple of analysis in the main text, then state that it has also been successfully applied to other data sets with details in supplementary material.*

—We have now shortened this section (from 57 lines in the former version to 36 lines in the current manuscript), in which we now minimally discuss biological insights from the results obtained with enhancers specific for hepatocytes, blood vessel endothelial cells, and macrophages, with enrichment analysis for other enhancers being mentioned briefly in the main text (page 7, lines 22–25) and shown in Figure EV4.

5. *Figure 1 caption C  "Numbers" should be replaced by "Proportion" as the figure does not provide actual numbers here.*

—We revised the legend of Figure 1C as suggested.

6. *In Table 1, one line about "Quality Control" should be added. Indeed, it is an important information that is not provided by ChIP-atlas but others. It would be fair to provide this information to the reader.*

—As suggested, we have now added a row regarding quality control in Table 1. We mention that, like ChIP-Atlas, GTRD also has no QC filtering, which has also been pointed in Table S4 of the ReMap paper (PMID: 29126285).

Corresponding Author Name: Shinya Oki

Manuscript Number: EMBOR-2018-46255V2